# Hurricanes and hashtags: Characterizing online collective attention for natural disasters

Michael V. Arnold[1,2,3]*, David Rushing Dewhurst[1,2,3,4], Thayer Alshaabi[1,2,3], Joshua R. Minot[1,2,3], Jane L. Adams[1,2,3], Christopher M. Danforth[1,2,3], Peter Sheridan Dodds[1,2,3]

**1** MassMutual Center of Excellence in Complex Systems and Data Science, University of Vermont, Burlington, Vermont, United States of America, **2** Computational Story Lab, Vermont Complex Systems Center, Burlington, Vermont, United States of America, **3** Department of Mathematics and Statistics, University of Vermont, Burlington, Vermont, United States of America, **4** MassMutual Data Science, Boston, Massachusetts, United States of America

* michael.arnold@uvm.edu

**Data Availability Statement:** All relevant data files are available on a public repository, hosted at github.com/mvarnold/Hurricane-Data-Repo.

## Abstract

We study collective attention paid towards hurricanes through the lens of *n*-grams on Twitter, a social media platform with global reach. Using hurricane name mentions as a proxy for awareness, we find that the exogenous temporal dynamics are remarkably similar across storms, but that overall collective attention varies widely even among storms causing comparable deaths and damage. We construct 'hurricane attention maps' and observe that hurricanes causing deaths on (or economic damage to) the continental United States generate substantially more attention in English language tweets than those that do not. We find that a hurricane's Saffir-Simpson wind scale category assignment is strongly associated with the amount of attention it receives. Higher category storms receive higher proportional increases of attention per proportional increases in number of deaths or dollars of damage, than lower category storms. The most damaging and deadly storms of the 2010s, Hurricanes Harvey and Maria, generated the most attention and were remembered the longest, respectively. On average, a category 5 storm receives 4.6 times more attention than a category 1 storm causing the same number of deaths and economic damage.

## Introduction

The collective understanding and memory of historic events shapes the common world views of societies. In a narrative economy, attention is a finite resource generating intense competition [1–9]. As commerce and communication shift to online platforms, so too has the narrative economy moved to the digital realm. In 2018, over $100 billion dollars were spent on internet advertising in the United States, nearly overtaking the $110 billion spent on traditional media advertising—about 1% of the US GDP [10]. Today, social media both facilitates and records an extraordinary percentage of the world's public communication [11, 12]. For

**Funding:** This work was supported by gifts from the Massachusetts Mutual Life Insurance Company and Google. Additionally, Massachusetts Mutual Life Insurance Company provided support in the form of salaries for author DRD, but did not have any additional role in the study design, data collection and analysis, decision to publish, or preparation of the manuscript. The specific roles of these authors are articulated in the 'author contributions' section.

**Competing interests:** I have read the journal's policy and the authors of this manuscript have the following competing interests: Author DRD's commercial affiliation with paid employment. Additionally, we have received funding from a commerical source: Massachusetts Mutual Life Insurance Company and Google. This does not alter our adherence to PLOS ONE policies on sharing data and materials.

computational social scientists, the migration of parts of the narrative economy to the web continues to present an immense opportunity, as the discipline becomes data-rich [13, 14].

Academics have become interested in narrative spreading around newsworthy events on social media platforms such as Twitter, as increasingly political fights for influence or narrative control are fought by actors as wide ranging from activists and police departments [15], to state censors suppressing discourse internally and state supported troll factories spreading divisive narratives internationally [16–21]. In 2019, the social media platform Twitter boasted over 145 million daily active users [22].

Quantifying the spread of narratives and the total attention commanded by them is a daunting task. Recent work has made progress in tracking the spread of quoted and modified phrases through the news cycle, and others have worked to identify actant-relationships and compile contextual story graphs from social media posts [3, 23]. In comparison, quantifying attention directed towards a topic, person or event is a somewhat easier task. Rather than identifying actors and identifying what they act on, as is the case for narrative attention, we can simply count mentions of an entity. Since increasing raw attention or number of mentions is often the zeroth order activity in public relations campaigns, quantifying the volume of attention, irrespective of the sentiment or narrative within which the attention is embedded, seems a natural first step [24–26].

An understanding of attention has typically focused on time dynamics as measured by the number of mentions in a given corpus, explaining either temporal decay of interest or heavy-tailed allocation of attention given to a spectrum of topics through some preferential attachment mechanism. [27–34]. Another group of studies have worked to classify attention time series from social media as either exogenous or endogenous to the system, modeling the functional form of collective attention decay, or determining if spreading crosses a critical threshold [35–38]. While these studies have typically focused on scientific works, patents, or cultural products such as movies, the rise of large social media datasets have enabled the investigation of a wider range of topics in online public discourse [39].

A broad spectrum of collective attention studies have been conducted using Twitter as a data source. Researchers have used Twitter data to indicate the likely outcome of elections by quantifying the collective attention directed toward political parties [40]. Other researchers have investigated the relationship between the dynamics of collective attention and event credibility, finding that "sustained, intermittent bursts of attention were found to be associated with lower levels of perceived credibility [41]."

In this study we examine the collective attention focused on hurricanes, using Twitter, which allows us to capture more natural speech intended for human readers as opposed to search terms. Twitter data has been used to measure shifts in collective attention surrounding exogenous events like earthquakes by looking for jumps in the Jensen-Shannon divergence between tweet rate distributions between days, or creating real-time earthquake detection using keyword based methods [42, 43].

Here, we use collective attention in a more narrow sense. Instead of looking for anomalous tweet rates, we study *n*-gram usage rates for hashtags and 2-grams associated with individual events. Specifically, we examine the usage rates of hashtags and 2-grams matching the case-insensitive pattern "`#hurricane*`" and "`hurricane *`", respectively. Natural disasters provide an ideal case study, since they are generally unexpected, producing the signature of an exogenous event. However, the volume of attention given to any particular hurricane varies widely across several orders of magnitude, as does the severity of the storm in terms of the lives lost and damages caused.

Prior efforts have examined the attention received by disasters by type and location, as measured by time devoted on American television news network coverage, and striking

discrepancies: for example, to have the same estimated probability of news coverage as a disaster in Europe, a disaster in Africa would need to cause 45 times as many deaths [44]. The same study found that in order to receive equivalent coverage to a deadly volcano, a flood would need to cause 674 times as many deaths, a drought 2,395 times as many, and a famine 38,920 times as many casualties.

Strong hurricanes are more likely to capture attention than weak hurricanes, and hurricanes impacting the continental United States capture much more attention than those failing to make landfall. To what degree does attention shrink when hurricanes make landfall outside of the continental US? The 2017 hurricane season is a particularly stark example, showing that for comparably powerful storms above category 4, those projected to make landfall over the continental United States were talked about nearly an order of magnitude more than Hurricane Maria, which impacted Puerto Rico, and two orders of magnitude more than Hurricane Jose, which never made landfall.

Given the attention received by some hurricanes so unbalanced, we must ask the question: Do government or humanitarian relief resources get dispersed with greater generosity for storms that capture public attention, or are these organizations insulated from popular attention? For the 2017 hurricane season, more money was spent more quickly to aid the victims of hurricanes Harvey and Irma than victims of Hurricane Maria, contributing to the significantly higher death toll and adverse public health outcomes in Puerto Rico [45]. While the attention and policies of government agencies are not usually dictated from Twitter, public attention certainly has some effect on the focus of agencies and allocation of government resources, and recently more attention has been focused on understanding the discourse on social media before, during, and after natural disasters [46–51]

We structure our paper as follows. In Results, we examine the spatial associations between hurricanes and the attention they receive, we compute and compare measures of total attention, maximum daily attention, and non-parametric measures of the rate of attention decay for the most damaging hurricanes in the past decade. We present conclusions in Concluding Remarks. Finally, we outline our methods and data sources, covering the collection of $n$-gram usage rate data in English tweets as well as data sources for hurricane locations and impacts.

## Materials and methods

### n-gram usage rates

We query the daily usage rate of hashtags referencing hurricanes are queried from a corpus of 1-gram—words or other single word-like constructs—usage rate time series, computed from approximately 10% of all posts ("tweets") from 2009 to 2019 collected from Twitter's "deca-hose" [52]. We define usage rate, $f$, as

$$f(t) = c_\tau(t) \Big/ \sum_{\tau' \in \mathcal{D}_t} c_{\tau'}(t),$$

with count, $c_\tau$, of a particular 1-gram divided is by the count of all 1-grams occurring on a given day, $\mathcal{D}_t$. The usage rates are based only on the usage rate of 1-grams observed in tweets classified as English by FastText, a language classification tool [53, 54]. We choose to focus on English tweets to study attention to North Atlantic storms, primarily because English is the most common language on Twitter. Additionally, US government agencies such as NOAA and FEMA compile estimates of hurricane impacts inside the US, a complementary dataset that we discuss below.

Our usage rate data set includes separate usage rates for 1-grams in "organic" tweets, tweets that are originally authored, as well as usage rates of 1-grams in all tweets (including retweets and quote tweets). More details about the parsing of the Twitter *n*-gram data set are available in [26].

For the purpose of studying attention, our usage rates are derived from the corpus with all tweets, including retweeted text, to better reflect not only the number of people tagging a storm, but also the number of people who decide the information contained therein was worth sharing.

We studied the usage rate of 1-grams exactly matching the form "`#hurricane*`", where `*` represents a storm's name. We also measured the usage rate of 2-grams matching the pattern "`hurricane *`" for each storm name. All string matching is case-insensitive. This choice is deliberately narrow, so that more broadly used hashtags do not inflate our measurement of attention associated with each storm. A broader measure is discussed in the S1 File and we show the two measures are strongly associated.

For the ten years covered by the HURDAT2 dataset overlapping with our Twitter dataset, there have been 75 storms reaching at least category 1 in the North Atlantic Basin. Within our 10% sample of tweets, we count over all storms a total of 1,824,842 hashtag usages within a year of each storm, and 3,643,411 instances of the matching 2-gram.

### Deaths, damages, and locations

To augment our usage rate data set, we downloaded data associated with all hurricanes in the North Atlantic basin from 2008 to 2019 from Wikipedia [55]. Included in the Wikipedia data are the damage estimates (US$) and deaths caused by each storm, as well as the dates of activity and areas effected. We also used the HURDAT2 data set containing the positions and various meteorological attributes of all North Atlantic hurricanes from 1900 to 2018 for the spatial component of this work [56]. For the time range overlapping with the Twitter derived data set, HURDAT2 has 3 hour resolution.

We note that we collected all data while complying with the terms and conditions of the respective websites.

## Results

### Hurricane attention maps

In Fig 1, we show hurricane positions as well as their hashtag usage rate timeseries with a time series indicating the usage rate of the hashtag of the form `#hurricane*`.

We plot the same hashtag usage rate time series below on both linear and logarithmic axes, as well as 2-gram usage rates. For clarity, we only include hurricanes reaching at least category 4.

The hurricane map tracks are meant to show the spatial dependence of attention given to hurricanes, while giving enough visual cues to connect locations along the path to the time the attention was observed. We generated the map shown in Fig 1 by filling in the polygon defined by the set of points lying at the end of a line segment of length proportional to the smoothed usage rate of the related hashtag, along the vector normal to the current velocity of the hurricane, and centered at the hurricane position at the given time. Maps of additional years are provided in the S9-S15 Figs in S1 File.

Our hashtag usage rate is at the day scale, while HURDAT has 3 hour resolution, so the wrapped attention volume is smoothed with a moving average with a window size of one day to avoid discontinuous jumps. This method obscures any sub-day scale resolution on the map, which could be related to the daily fluctuation of tweet volume as well as varying interest in the

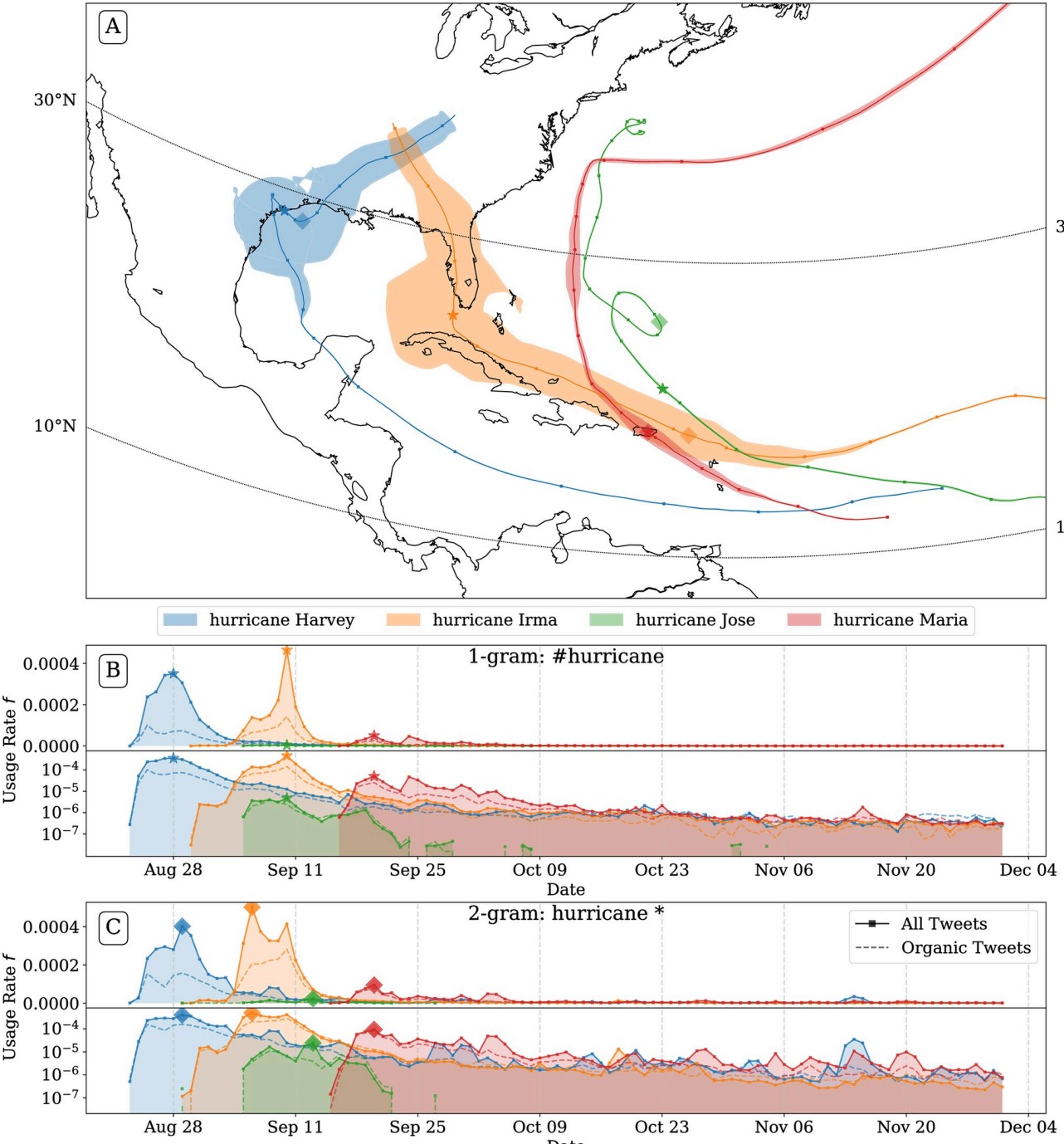

**Fig 1. Hashtag attention map and usage rate time series.** For 1-grams matching the case-insensitive pattern "`#hurricane*`" for all four hurricanes reaching at least category 4 in the 2017 hurricane season. Markers along the hurricane trajectory indicate the National Oceanic and Atmospheric Administration (NOAA) reported position for every day at noon UTC. On the map, the smoothed rate of hashtag usage is wrapped in an envelope around the hurricane trajectory in panel A, showing the spatial dependence of attention on Twitter. In the lower two plots, panels B and C, we show the usage rates for hashtags and 2-grams matching `hurricane*` in English language tweets on linear and logarithmic scales. Usage rates within all tweets are indicated with a solid line, while usage rates in 'organic' tweets (tweets that are not retweets), are represented by a dashed line. The day of maximum attention on Twitter is marked with a star or a diamond for hashtags or 2-grams, respectively. Generally, hurricanes making landfall on the continental United States received greater attention than those not making landfall. The hashtag usage rate for hurricanes Harvey and Irma at their maximum were approximately an order of magnitude larger than the maximum hashtag usage corresponding to Hurricane Maria, and two orders of magnitude larger than Hurricane Jose.

hurricanes. While we lose some granularity using daily usage rates, the decays in attention are spread out over days and weeks for smaller storms, and months for larger storms. Daily resolution is sufficient to capture the longer decays in attention, which are our primary interest.

Examining the map, we can see the minimal attention paid to Hurricane Harvey as it traveled across the Caribbean sea and made landfall in Mexico. It is only after crossing the Gulf of Mexico that the hashtag registered on our instrument, and only when it was about to make landfall over Texas did the hashtag usage rate approach its maximum rate, approximately 3 of every 10,000 1-grams in English tweets. It appears that the devastation wrought by Harvey primed hurricane-related conversation, as the next hurricane, Irma was talked about long before it made landfall. While Irma was talked about with a similar usage rate as Harvey as it impacted Puerto Rico, Hispaniola, and Cuba, it spiked while making landfall in the Florida keys.

Comparing the attention generated by the previous two storms, Hurricane Maria generated substantially less hashtag usage. The peak of its attention gathered as it made landfall over Puerto Rico as a category 4 storm, with less than a fifth of the attention as the hurricanes making landfall on the US. Part of the reason may be due the affected area being Spanish speaking, while our hashtag usage measurement only counts occurrences in English tweets. We find that usage rates of the 2-gram "`huracán maria`" in Spanish tweets were also lower than the usage rates for "`huracán irma`", but comparable to those for "`huracán harvey`." See S2 and S3 Figs in S1 File to compare top hurricane related 2-gram time series for the 2017 hurricane season in English and Spanish.

Another potential contributing factor for the low volume of Hurricane Maria tweets could be that Puerto Rico's electric grid was destroyed and 95% of cell towers were down in the aftermath of the storm, making it impossible for those directly affected to communicate about the storm [57]. Unfortunately, due to Twitter's usage norms in this time period, we do not have locations for the vast majority of tweets. The number of people affected by the storms could also help explain the different levels of attention, as both Hurricane Harvey and Irma affected 19 million people, while Maria affected about 4 million [58].

## Hurricane attention comparison

To compare the variation in attention received by different storms, we combined measurements of the hashtag usage rate with deaths and damages caused by each storm from 2009 to 2019.

In Fig 2, we show radar plots (radial, categorical charts) comparing six measurements of impact and attention for each of the eight most damaging hurricanes in the time period of study [59]. S8 Table in S1 File shows the raw measured values for the most damaging hurricanes in this period.

Included measurements are:

- Max Usage Rate—peak attention on any single day

- Integrated Usage Rate—total attention over the entire hurricane season

- Quantile 0.9: $Q_{0.9}$—days to 90% attention

- Quantile 0.99: $Q_{0.99}$—days to 99% attention

- Damage—total damage caused by the storm in US dollars

- Deaths—total deaths associated with the storm (both direct and indirect)

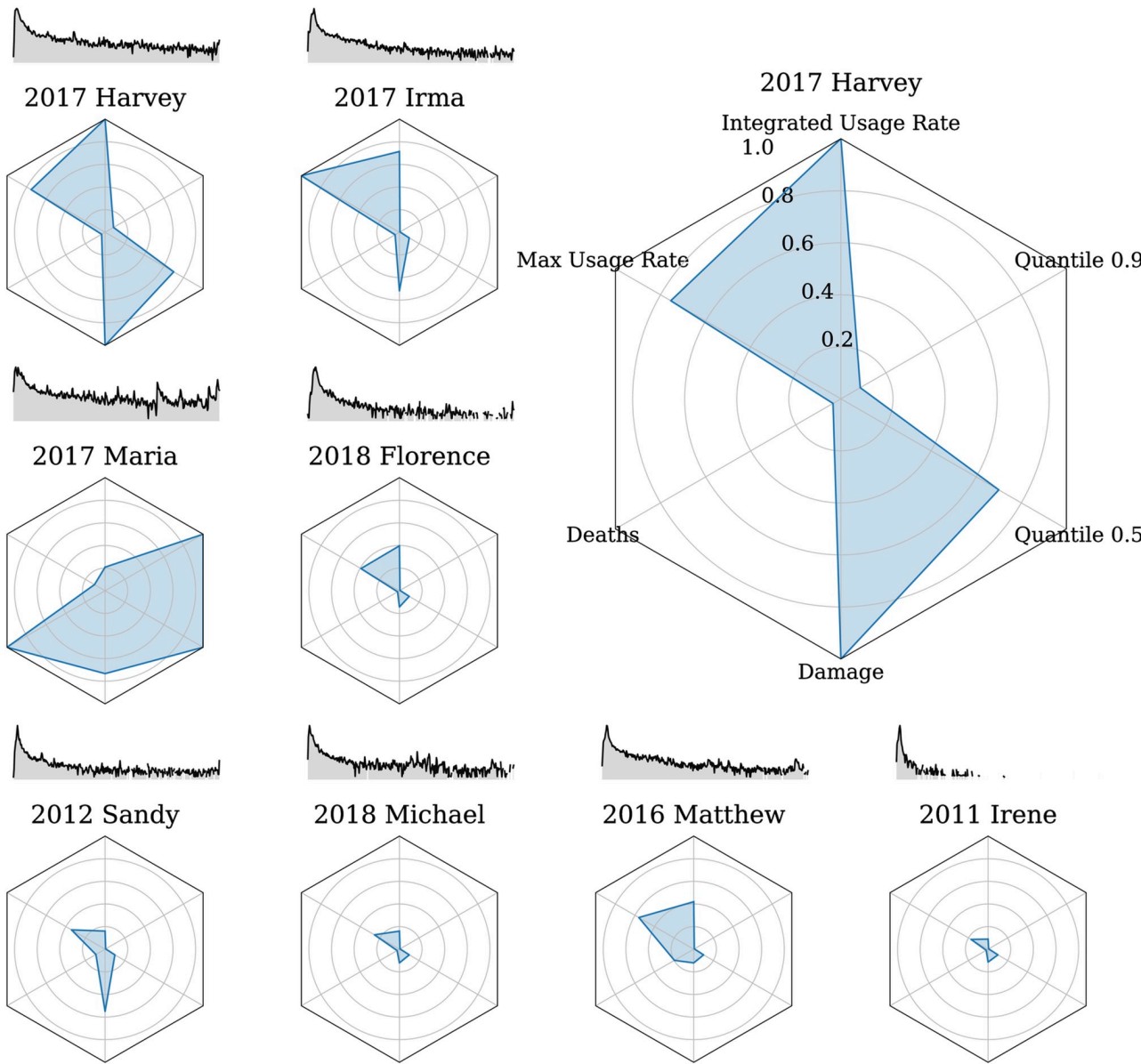

**Fig 2. Radar plots comparing the eight most monetarily damaging hurricanes in the North Atlantic basin from 2009 to 2018.** For each plot, starting at the top position and rotating clockwise the measures are: the sum of usage rate of the hashtag, the number of days to reach 90% and 50% of the total attention received during that season, the total cost in dollars attributed to damage caused by the hurricane (in its year), the number of deaths attributed to the hurricane, and maximum usage rate of the hashtag during the year of interest. All measurements are normalized to the maximum value achieved by any hurricane. Hurricane Harvey was the most talked about hurricane, as well as the most damaging. Hurricane Irma was the most talked about on any single day. Hurricane Maria caused the most deaths, and had the longest attention half-life of all measured hurricanes. Raw values for this figure are shown in S8 Table in S1 File. Hashtag usage rate spark lines above each radar plot are normalized to show the common decay shape, and can not be compared to evaluate *relative* volume, and are shown on a log scale.

The relative magnitude of each quantity is shown as a fraction of the maximum value for any storm in the study. The quantile values are non-parametric measurements of the attention time scale—comparable to half-lives but without the assumption of an exponential decay. Some storms receive significant interest months after they pass, usually related to the recovery efforts. Spark lines above each plot show the attention time series for the year after each storm, as measured by the log usage rate, but do not convey relative scale.

The three most damaging storms, Hurricanes Harvey, Maria, and Irma, all destroyed tens of billions of dollars of property. Storms in Fig 2 are ordered by damage, with the least damaging being Hurricane Irene in 2011, which still destroyed an estimated $14 billion in property.

The most deadly North Atlantic hurricane in the past decade was Hurricane Maria, killing over 3000 people over the course of the extended disaster. The next most deadly storms were Hurricanes Matthew, Sandy, Irma, and Harvey, all killing at least 100 people. Among the storms shown in the Fig 2, Hurricanes Florence and Irene were the least deadly, causing 58 and 57 deaths, respectively.

The highest hashtag usage rate on a single day was associated with Hurricane Irma, reaching $\max f_\tau = 4.6 \times 10^{-4}$, or 4.6 of every 10,000 1-grams, as the storm made landfall over the Florida Keys. Other storms reached comparable single day usage rates, such as Hurricanes Harvey and Matthew, reaching $\max f = 3.5 \times 10^{-4}$ and $\max f = 2.6 \times 10^{-4}$, respectively. Within the top eight most damaging storms, the hashtag associated with Hurricane Maria had the lowest maximum usage rate. The hashtag "#hurricanemaria" appeared only five times for every 100,000 1-grams as Maria made landfall in Puerto Rico.

The highest integrated hashtag usage rate was associated with Hurricane Harvey, followed by Hurricanes Irma, Matthew, and Florence. The integrated hashtag usage rate for "#hurricaneharvey", $I = 2.3 \times 10^{-3}$. Hashtags associated with Hurricanes Sandy and Irene had the total attention, with $I = 3.7 \times 10^{-4}$ and $I = 2.0 \times 10^{-4}$, respectively.

Due to the extended crisis in the aftermath of Hurricane Maria, the hashtag continued to be used at relatively high volumes even a year after the storm had passed, leading to much larger value for $Q_{0.9}$ of 175 days [60, 61]. Typical values for $Q_{0.9}$ were around 1–4 days, with more prolonged and damaging storms like Harvey in 2017 taking 15 days to reach 90% total attention. In comparison no other storm took longer than 100 days to reach this benchmark. We chose the longer term attention timescale benchmark, $Q_{0.99}$, to describe how long until nearly all storm focused attention has passed. We observe the hashtag associated with Hurricane Maria is the largest for this measurement as well, with $Q_{0.99}$ of 363 days, which should be interpreted as attention not dying away within a year, since we truncate the timeseries after one year. Hurricane Michael, Sandy, and Harvey also have triple digit values for $Q_{0.99}$, as they continued to be talked about, albeit at much lower levels than their peak. Other storms quickly lose attention, such as Hurricane Irene, which took only 12 days to reach 99% total attention.

We observed variation in the overall radar plot shape. More recent storms have been more damaging and deadly, and we find higher measures of total attention and attention decay. A number of storms like Sandy, Michael, and Matthew have relatively higher values for both maximum usage rate and number of days to reach 99% total attention. While there is significant variation in the magnitude of these measurements, the essential exogenous shape of the hashtag usage rate timeseries, $f$, is consistent. We fit a bi-exponential decay model to further quantify how quickly attention decreases, and present the fitted half-lives in S9 and S10 Tables in S1 File.

## Attention and impact regressions by category

We next explore the associations between damage, deaths, and attention given to hurricanes. In Fig 3, we show the scaling relationship between attention and impacts for each category storm on the Saffir-Simpson wind scale [62]. The scale assigns a hurricane a category from one to five based on the sustained wind speed. Importantly, this category is often the descriptor used by metrologists to communicate the severity of the storm to the public. For the regression, we assign a storm the maximum observed category. Each sub-panel plots the integrated usage rate, $I = \Sigma_t f(t)$ for hashtag or 2-gram $\tau$, against a measure of storm impact, where $t$ runs

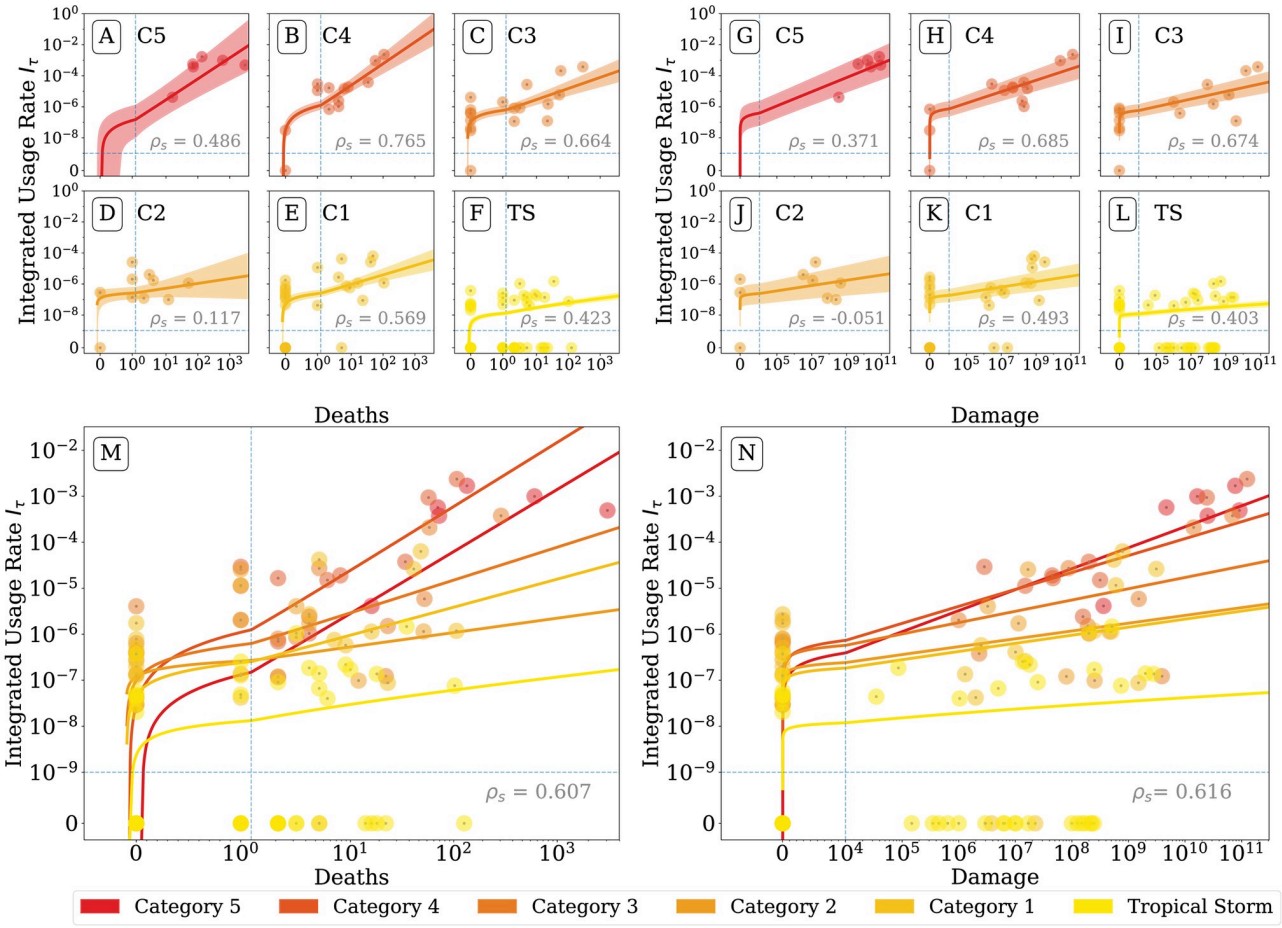

**Fig 3. Scatter plots for integrated hashtag usage rate versus the deaths and damages caused by each storm.** There is a clear positive association between the total attention represented by hashtags and the impacts of these storms. We reported Spearman's rho, $\rho_s$, in the top left corner of each plot. While for some categories, there is little evidence for a positive association, for the entire dataset $\rho_s \sim 0.54$. We perform a Bayesian linear regression for each category storm between the $\log I$ and log impacts. We show the mean model, along with the credible interval within a standard deviation of the mean model. We use hybrid axis with logarithmic scaling for most horizontal and vertical values and linear scaling near zero, in order to show storms that caused zero deaths or damages, as well as storms for which we measured a hashtag usage rate of zero. Changes in axis scaling occur at the blue dashed lines. Generally, more powerful storms received more attention, higher category storms received more attention even when causing minimal damage, and high category storms had a higher regression slope. These results suggest that for powerful storms, a given increase in impact was associated with a larger increase in attention. While for category one storms a 10-fold increase in deaths is associated with a four-fold increase in attention, for category five hurricanes, this same 10-fold increase in attention is associated with a 25-fold increase in attention.

over an index of the 365 days after each storm began. $I$ is chosen as a measure of total attention given to the storm during its respective hurricane season, which can be compared across years since it is already normalized to the total volume of conversation on Twitter. Color represents the maximum category storm reached, and the smaller subplots are breakout panels for each category. We include Spearman's $\rho$, a non-parametric measure of rank correlation, in each panel.

We perform linear regressions on storms in each category separately, a choice that models the attention received by different category storms as separate processes. With models in Regression Models for Impacts, Impact Interactions and Hurricane Category, we separately consider attention as a singular process where we account for the hurricane's maximum category rating using an explicit indicator variable.

**Model choice and fitting procedure.** For each category and each impact, we model total attention as

$$\log_{10} I = a_0 + a_{\text{impact}} X_{\text{impact}} + \varepsilon_\tau, \tag{1}$$

where $X_{\text{impact}}$ is either $\log_{10}$ deaths or $\log_{10}$ damages caused by each storm. We use a logarithmic model both to capture the scaling relationships between impacts and attention and to inform on the relative changes in attention associated with storm impacts. We offset $I$ by $10^{-8}$ and the log impacts, $X_{\text{impact}}$ by \$10,000 and 0.1 deaths, respectively to avoid divergent log data where observed values are equal to zero.

We set a zero-centered normal prior on the slope of the regression model as $a_1 \sim$ **normal** $(0, 1)$. We set a normal prior on the intercept of the model with mean equal to $\log_{10} I = -8$, the minimum value of the offset added to $I$. We did not have strong beliefs about the likely precision of $a_0$ since it was not *a priori* clear how much attention would be paid to hurricanes with very little associated monetary damage or few deaths. We thus set a weak hyper-prior on the precision of $a_0$, $\tau \sim$ **gamma**$(3, 1)$; the intercept of the regression is distributed as $a_0 \sim$ **normal** $(-8, \tau^{-1})$.

We found regression coefficients by sampling with the No-U-Turn-Sampler (NUTS), using 8 chains with 2000 draws each after 1000 steps of burn-in [63]. Our models converged, with the Gelman-Rubin statistic, $\hat{R}$, never exceeding 1.004 for any model in the 12 models fit.

**Model posteriors and discussion.** In Fig 3, we show the fitted regressions for each category. The size of the impact and attention variables vary over many orders of magnitude, but also include zero values, corresponding to storms that cause no deaths or damage, or had zero usage of the hashtag associated with their name during the year the storm was active. Note that it should not be surprising that tropical storms appear to receive less attention via our hashtag usage rate measurement, since they never officially become hurricanes, and thus many of the tropical storm hashtags have an integrated usage rate, $I = 0$.

To display all data, we use symmetric log axes: logarithmic for large values and linear for small values. We indicate the switch point from linear to log space axis as blue dotted lines. This choice of axes causes the linear regressions on the log transformed data to appear curved for small values.

In each of the small subplots of Fig 3, we show the $1\sigma$ credible interval for the model as a band around the mean regression model. The credible interval is noticeably wider for category five storms, which is reasonable given there are only six storms reaching this category. Generally the mean regression lines are ordered such that higher category storms are receiving more attention than lower category storms. The slopes of the regressions are also higher for higher category storms. However, to better understand the models, we need to compare the model parameters individually.

In Fig 4 we provide posterior distributions for model parameters, which show that, as expected, more intense storms receive more attention per unit of log impact than weaker storms. For category five storms, we find a mean regression co-efficient of $a_{\text{deaths}} = 1.35 \pm 0.39$, using the format $\mu \pm \sigma$ where $\mu$ is the mean and $\sigma$ is the standard deviation, while for category one storms we find a mean regression co-efficient of $a_{\text{deaths}} = 0.61 \pm 0.18$. For a Table of mean parameter values, see S1 Table in S1 File.

Looking at associations between log damages and log attention we find $a_{\text{deaths}} = 0.46 \pm 0.07$ for category 5 storms, while for category one storms we find $a_{\text{deaths}} = 0.17 \pm 0.05$.

To interpret the regression coefficients, $a_{\text{impact}}$, as representing proportional increases in attention per proportional increase in impact, we exponentiate the coefficient. Thus, our model shows a 10-fold increase in deaths for a category 5 storm is associated with a 22-fold

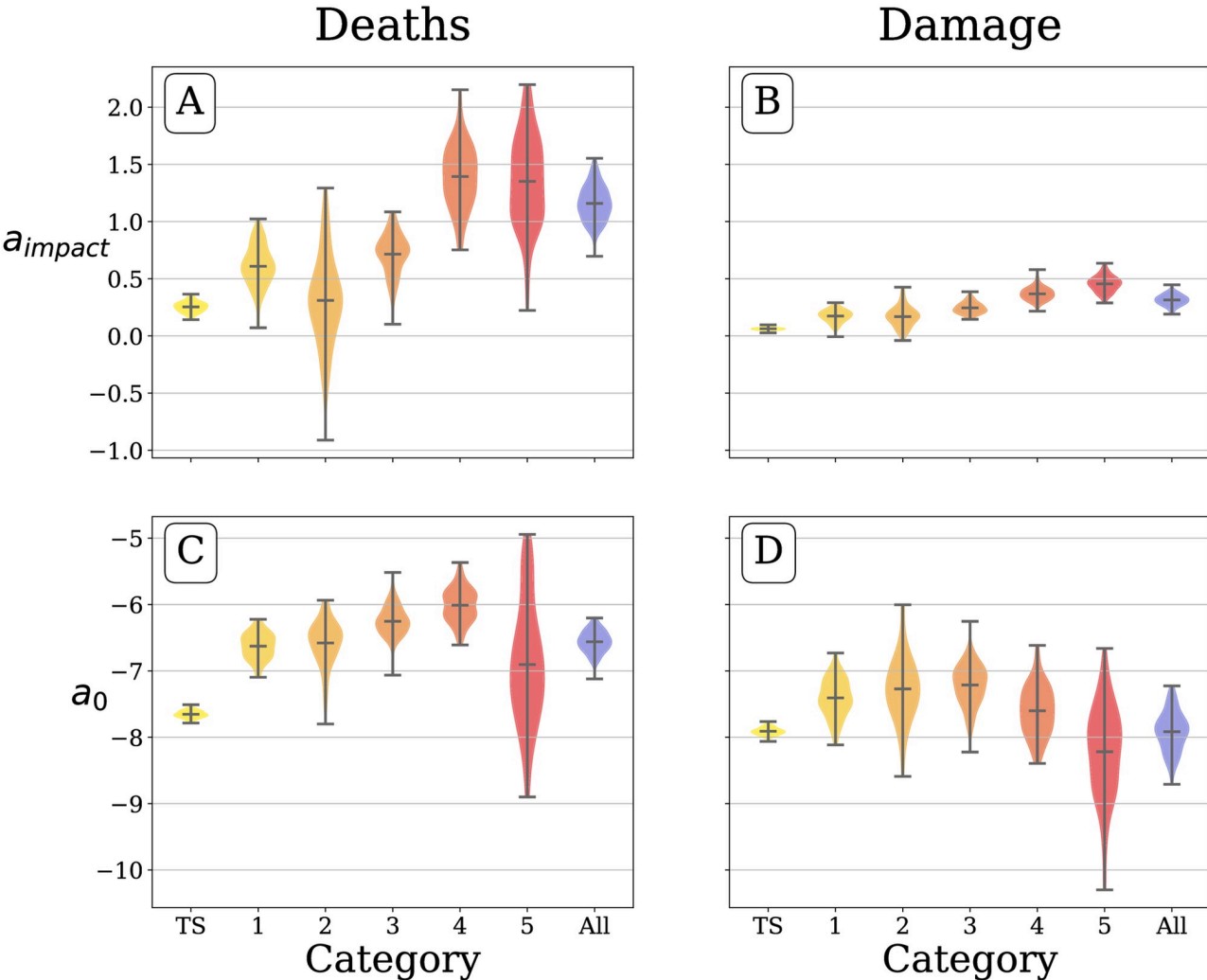

**Fig 4. Posterior distributions of regression parameters.** For the model $\log_{10} I \sim a_0 + a_1 X_i$, where $X_i$ is either the log number of deaths (A and C) or log damages in dollars associated with the storm (B and D), and $\log_{10} I$ is the log integrated hashtag usage rate. The trend in regression coefficients for association between the log attention and log deaths suggests that higher category storms receive more attention per unit impact, while the trend of intercepts shows increasing baseline attention for a hypothetical minimally disruptive storm causing exactly $1 in damages or one death. For regression coefficients relating log attention to log damages, Category 4 and 5 storms receive more attention per unit increase in log damages than lower category storms. However, the coefficients are smaller in magnitude due to damages varying across 7 orders of magnitude, as compared to deaths varying over 4 orders of magnitude. There is a larger uncertainty for the category 5 intercept values, as only 6 storms of this intensity formed between 2009 and 2019 in the Atlantic basin. At the right of each plot, we show the coefficients for the model fit for all hurricanes (blue violin), excluding tropical storms. Above each category, we show the value of the mean posterior distribution for each parameter. For a Table of mean parameter values, see S1 Table in S1 File.

increase in attention, while for a category 1 storm the same 10-fold increase in deaths is associated with a 4-fold increase in attention.

The intercepts, $a_0$, for higher category storms tend to be larger, meaning that for a theoretical minimally disruptive storm causing exactly $1 of damages or one death, a powerful storm would be talked about more, as shown in Fig 4. We believe this trend could continue for category 5 storms, but we have observed only $n = 6$ such storms for the duration of our attention dataset. We interpret the intercepts as indications of how much attention low-impact storms receive on average.

In Fig 4, we fit another regression model on all hurricanes examining log deaths and log attention. We find a 10-fold increase in deaths is associated with a 14-fold increase in attention, since the mean value of $\bar{a}_{deaths} = 1.16 \pm 0.15$ For damages, coefficients tend to be lower than those for deaths: $\bar{a}_{damage} = 0.31 \pm 0.05$. We interpret this coefficient as a 10-fold increase in damage being associated with no more than a 2-fold increase in attention.

## Regression models for impacts, impact interactions and hurricane category

In order to better understand the scaling of attention with hurricane impacts, we fit a number of models on the log transformed data. We applied the same offsets as in the previous section to avoid non-finite log transformed data. We exclude tropical storms, since their attention is not captured in same way as our string matching for hurricanes.

**Regression 1.** We fit the regression model,

$$\log_{10}I = a_0 + a_{\text{death}}X_{\text{death}} + a_{\text{damage}}X_{\text{damage}} + \varepsilon, \qquad (2)$$

where both predictors $X$ are log impacts, which we refer to as regression 1. The regression coefficients can be interpreted as the increase in log attention received for every unit increase in log impact. Likewise, the intercept can be interpreted as the expected attention for a minimally damaging storm causing one death and \$1 of damage. This model is distinguished from the previous section by including both log impacts in a single model, while not including an interaction term as later models will.

We set priors for the model as shown in S2 Table in S1 File. We chose the intercept, $a_0 \sim$ **normal**$(-8, 3)$, to be centered around -8, approximately the lowest usage rate captured in our data, as we guess storms causing 1 death and \$1 worth of damage are talked about relatively little, but wish to allow a wide range of uncertainty spanning a few orders of magnitude. We chose the priors for the regression coefficients, $a_{death} \sim$ **normal**$(0, 1)$ and $a_{damage} \sim$ **normal** $(0, 1)$, to be weakly informative and centered around zero, as to not bias towards any association. We sampled the coefficients' posterior distributions using NUTS, using 8 chains with 2000 draws each, after 500 steps of burn-in [63]. We found the model converged, with the maximum value of $\hat{R} = 1.000$.

We show the posterior distributions of model parameters for regression one in Panel A of Fig 5, which have a positive scaling between both deaths and damages, and the amount of attention commanded by the storm, as measured by the log hashtag usage rate. We interpret the mean value of $a_0 = -7.57 \pm 0.5$ for the regression constant as the expected log hashtag usage rate for a minimally destructive storm, i.e., that in English tweets, the hashtag usage rate would integrate to $10^{-7.57}$ over the season. We provide summary statistics in S3 Table in S1 File.

At first glance, this level of attention seems remarkably low: if occurring all in a single day, this is little more than 1 usage for every 100 million 1-grams. The most devastating storms can have integrated usage rates of $I = 2.3 \times 10^{-3}$, five orders of magnitude more attention than our regression constant. However, the least impactful storms affect relatively few people, while the most destructive storms significantly disrupt the lives of tens of millions, so the differences in the scale of total hashtag usage rate are not unreasonable. See S8 Table in S1 File for measured values corresponding to each storm.

We find $a_{\text{death}} \simeq 0.49$ and $a_{\text{damage}} \simeq 0.24$. Because $10^{0.24} \simeq 1.7$, considering the results in linear space, a 10-fold increase in damages is associated with a 1.7-fold increase in hashtag usage rates, while a 10-fold increase in deaths is associated with a 3-fold increase.

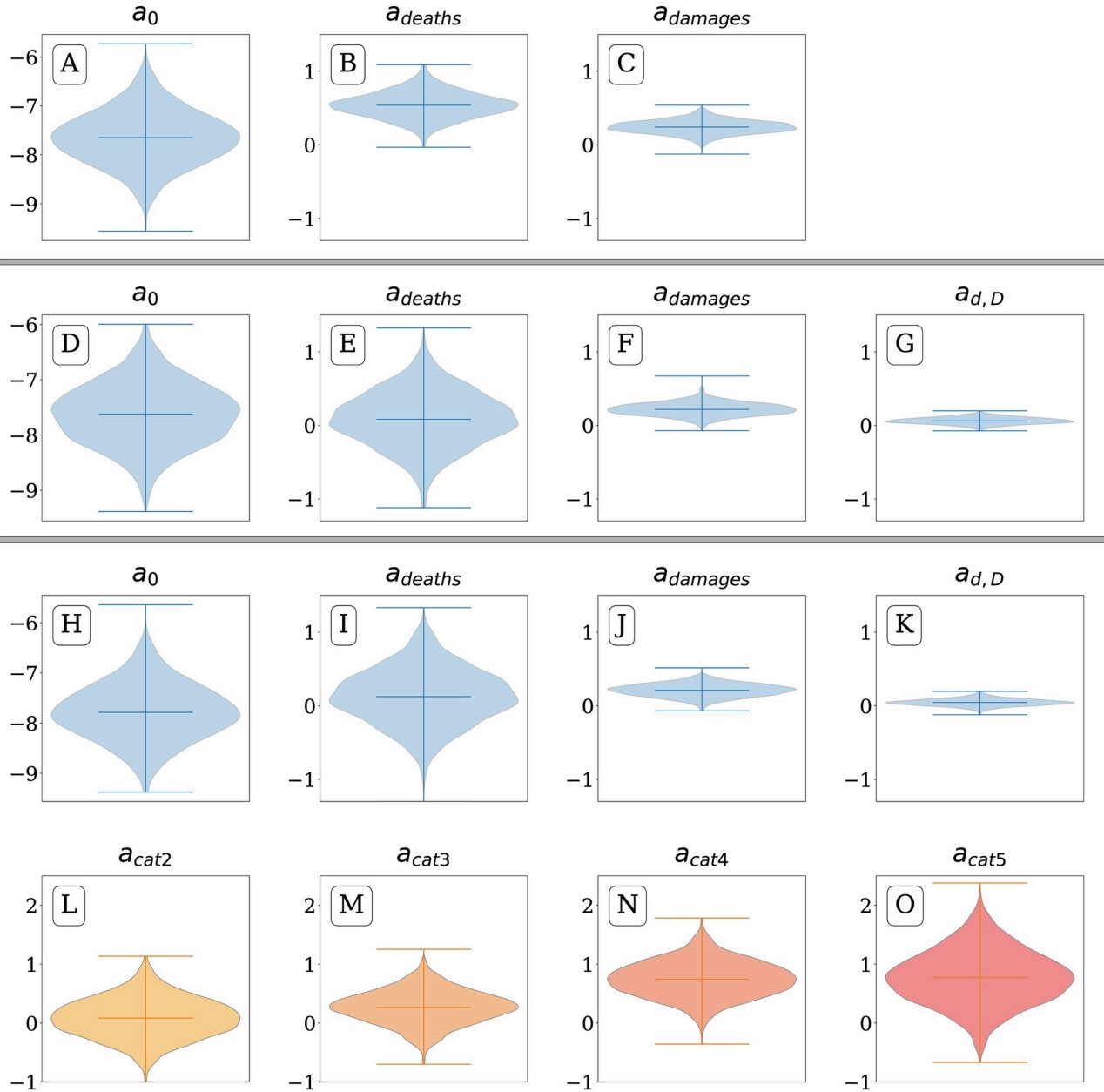

**Fig 5. Parameter distributions for models 1, 2 and 3.** Plots A–C show posterior distributions for regression 1, plots D–G show distributions for regression 2, which includes the addition of an interaction term, and plots H–O showing distribution for regression 3, which includes indicators variables for hurricane categories two through five. The addition of the interaction term, $a_{d,D}$ increases posterior variance for $a_{\text{deaths}}$ as well as reducing its mean from $a_{\text{deaths}} = 0.49$ in regression 1 to $a_{\text{deaths}} = 0.05$ in regression 2 and $a_{\text{deaths}} = 0.12$ in regression 3, suggesting that while the number of deaths is associated with increased attention, attention response is primed by destruction. Additionally, the hurricane category indicator variables in regression 3 show the progressive increase in attention given to higher category storms compared to category 1 hurricanes.

**Regression 2.** For the second regression, an interaction term was introduced between the log number of deaths and the log damages,

$$\log_{10}I = a_0 + a_{\text{death}}X_{\text{death}} + a_{\text{damage}}X_{\text{damage}} + a_{d,D}X_{\text{death}}X_{\text{damage}} + \varepsilon. \tag{3}$$

Prior distributions for the intercept and main effect coefficients are unchanged from regression 1. We set the prior distribution for the interaction coefficient to be $a_{d,D} \sim \textbf{normal}(0, 1)$, a standard weakly informative prior for regression coefficients. All priors are shown in S4 Table in S1 File. We used identical fitting procedures as above, and found the models converged with a maximum value of $\hat{R} = 1.0001$.

Here, the intercept is largely the same as the simplest regression model. Interpreting $a_{\text{death}}$ as the conditional relationship between log usage rate and log deaths when total damage is \$1, the $a_{\text{death}} = 0.05$ implies that for a 10-fold increase in deaths is associated with a 1.12-fold increase in hashtag usage rate, though the standard error includes zero. Similarly, $a_{\text{damage}} = 0.22$ implies a 10-fold increase in damage is associated with a 1.6-fold increase in hashtag usage rate. Finally, the interaction coefficient $a_{d,D}$ is small, but positive: a 10-fold increase in $X_{\text{death}} X_{\text{damage}}$ is associated with a 1.14-fold increase in hashtag usage rate. Notably, the inclusion of the interaction term significantly reduces the regression coefficient associated with deaths, while the coefficient associated with damage is largely unchanged. We provide summary statistics in S5 Table in S1 File.

This provides evidence that storms that cause a large number of deaths and damages are associated with higher volumes of attention, while a storm causing a large number of deaths but relatively less damage will attract much less attention for Twitter users. One possible explanation for this is that attention is primarily driven by those directly affected, while the Twitter users are not evenly distributed throughout the population. Wealthy people are over-represented among Twitter users, and thus hurricanes that affect capital-poor regions also affect few Twitter users. Second, we performed this regression on data from English language tweets, so the attention paid to storms effecting Spanish speaking regions is an underestimate.

**Regression 3.** To better understand the effect of hurricane category on attention, we performed a regression including this categorical variable, modeled as

$$\log_{10} I = a_0 + a_{\text{death}} X_{\text{death}} + a_{\text{damage}} X_{\text{damage}} + a_{d,D} X_{\text{death}} X_{\text{damage}} + \sum_j a_{C_j} X_{C_j} + \varepsilon, \qquad (4)$$

where the index $j$ runs from 2 to 5. We did not include a variable for category 1 hurricanes to avoid issues of multi-collinearity. Fitting procedures were identical to above, and we found the model converged with the max value of $\hat{R} = 1.0003$.

We did not change priors for the model coefficients from above for existing parameters, and we set the coefficients for category indicator variables to a weakly informative prior, $a_{C_i} \sim \textbf{normal}(0, 1)$. A Table of priors is shown in S6 Table in S1 File. Since we have included our hurricane categories, the interpretation of the intercept $a_0$ is now the expected log integrated hashtag usage rate $I$ for a category one hurricane, which causes one death and \$1 of damage. The value is similar to the other regression models. Effect sizes for $a_{\text{damage}}$ and $a_{d,D}$ are reduced in magnitude slightly compared to the preceding regression.

As measured by the integrated hashtag usage rate, compared to a category 1 storm causing the same deaths and damages, hurricanes in:

- category 2 receive 1.14 times more attention,

- category 3 receive 1.5 times more attention,

- category 4 receive 5.6 times more attention,

- and category 5 receive 4.6 times more attention.

We show the posterior distributions for regression three in Panel C of Fig 5, show summary statistics in S7 Table in S1 File, and summarize the results of all models in Table 1.

**Table 1. Regression summaries.** For each model presented in the paper. SubTable A refers to the regressions by category, while subTable B refers to the later sequential regression models. Each impact variable is presented as the expected increase in attention associated with a 10-fold increase in the variable of interest. For categorical variables we report the expected multiplier for the given hurricane category over a Cat 1 storm. The mean of the fitted posterior regression parameters are provided for the reader in the Appendix in S3, S5 and S7 Tables in S1 File.

**A**

| For a 10-fold ↑ in $X_{\text{impact}}$ | | TS | Cat 1 | Cat 2 | Cat 3 | Cat 4 | Cat 5 |
|---|---|---|---|---|---|---|---|
| | $X_{\text{deaths}}$ | 1.8 | 4.1 | 2.6 | 5.0 | 26.9 | 24.5 |
| | $X_{\text{damage}}$ | 1.3 | 1.9 | 1.7 | 2.2 | 3.0 | 2.9 |

**B**

| For a 10-fold ↑ in $X_{\text{impact}}$ | Regression 1 | $X_{\text{deaths}}$ | $X_{\text{damage}}$ | | |
|---|---|---|---|---|---|
| | | 3.1 | 1.7 | | |
| | Regression 2 | $X_{\text{deaths}}$ | $X_{\text{damage}}$ | $X_{\text{deaths}} * X_{\text{damage}}$ | |
| | | 1.1 | 1.7 | 1.1 | |
| | Regression 3 | $X_{\text{deaths}}$ | $X_{\text{damage}}$ | $X_{\text{deaths}} * X_{\text{damage}}$ | |
| | | 1.2 | 1.58 | 1.12 | |
| Multiplier over Cat 1 | | Cat 2 | Cat 3 | Cat 4 | Cat 5 |
| | | 1.14 | 1.5 | 5.6 | 4.6 |

## Concluding remarks

We have explored the attention given to hurricanes as measured by the hashtag and 2-gram usage rate. We quantify the relative volume of attention time series for major storms. We find evidence that not only are more powerful—higher maximum category rating—storms talked about more than weaker storms, but they are talked about more when they inflict the same amount of damage or take the same number of lives. Further, different attention scaling relationships exist for different category storms. For the most destructive storms, we demonstrate that a 10-fold increase in deaths is associated with a 25-fold increase in attention, while for weaker storms the same proportional increase in deaths would lead to only a four-fold increase in attention on average.

How people outside of the government agencies and non-governmental organizations (NGOs) tasked with responding to natural disasters perceive the importance of disasters have real-world consequences [64, 65]. We hypothesize that monetary donations to NGOs that assist with hurricane disaster relief efforts are strongly associated with the amount of attention attracted by the hurricane. If this is true, it could be advantageous for NGOs to prospect for financial contributions within the narrow time window when collective attention is focused most strongly on a storm [66]. It is also possible that the speed and scale of governmental relief programs are influenced by popular attention paid to storms, and previous work has shown that relief has been inequiTable in the past [45]. Future work could compare the quantities of non-profit and governmental assistance with attention volume.

While the users of Twitter are certainly not representative of the world, or even English speakers, measuring the text they generate approaches measurement of the population at large, at least more-so than published books or edited newspaper columns [67–71]. The digital signatures left behind by our collective online presence offers rich data for observational studies of everyday language with unprecedented time resolution. Of course, many tweets referencing hurricanes are authored by journalists or news organizations and future efforts could attempt to disentangle the various motivations contributing to the overall usage rate of hashtags and other $n$-grams.

Another limitation of our work, particularly relevant to any geospatial findings, is that we only consider tweets classified as English. While the density of English speakers closely mirrors the population density for much of the United States, we observe much lower usage rates for the English language hashtags and 2-grams over predominately Spanish speaking areas. While different populations may use different *n*-grams to reference the same storm, for the purposes of our study we have focused only on the English-speaking population of Twitter.

Future work could consider how to better quantify the total fraction of conversation of Twitter focused on a storm or event of interest. Our current method only includes counts for individual *n*-grams, which we believe acts as a proxy of total attention, but almost certainly underestimates the total fraction of text devoted to discussing a topic. Hashtag co-occurrence network-based methods could help to identify the most prominent hashtags associated with a given storm, or any event of interest, and to classify tweets as relevant. Examining properties of this network changing in time, such at the integrated usage rate of all significant hashtags within one degree could give a more unbiased view of the total attention surrounding the hurricane than our current method. Other dynamics of hurricanes could be explored in this way, perhaps by encoding Jenson-Shannon Divergence shifts between hashtags as a node attribute [72], or more simply how the most frequently used hashtags in this ego network change in rank over time, as different phases of the storm occur. With better data coverage for infrequently used hashtags, the effect of new storms on the attention paid to historical storms could be studied using a measure similar to view flow [73]. Authors of previous works studying the effectiveness of NGO hashtag usage following natural disasters could exploit these network based methods [74].

## Supporting information

**S1 File.**
(PDF)

## Acknowledgments

The authors are grateful for the computing resources provided by the Vermont Advanced Computing Core.

## Author Contributions

**Conceptualization:** Michael V. Arnold, David Rushing Dewhurst, Thayer Alshaabi, Joshua R. Minot, Jane L. Adams, Peter Sheridan Dodds.

**Data curation:** Michael V. Arnold.

**Formal analysis:** Michael V. Arnold.

**Investigation:** Michael V. Arnold.

**Software:** Michael V. Arnold.

**Supervision:** Christopher M. Danforth, Peter Sheridan Dodds.

**Visualization:** Michael V. Arnold, Thayer Alshaabi, Jane L. Adams.

**Writing – original draft:** Michael V. Arnold.

**Writing – review & editing:** David Rushing Dewhurst, Thayer Alshaabi, Christopher M. Danforth, Peter Sheridan Dodds.

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
