## [Decision Letter · Decision Letter 0]

13 Apr 2021

PONE-D-21-06464

Hurricanes and hashtags: Characterizing online collective attention for natural disasters

PLOS ONE

Dear Dr. Arnold,

Thank you for submitting your manuscript to PLOS ONE. After careful consideration, we feel that it has merit but does not fully meet PLOS ONE’s publication criteria as it currently stands. Therefore, we invite you to submit a revised version of the manuscript that addresses the points raised during the review process.

The reviewers are happy with the revision. The only one thing Reviewer 1 wants to check is the usage of non-accented version of the word. The paper can be accepted if authors can quickly check this and provide a brief response. 

We look forward to receiving your revised manuscript.

Kind regards,

Yong-Yeol Ahn, Ph.D.

Academic Editor

PLOS ONE

Journal Requirements:

"This work was supported by gifts from the Massachusetts Mutual Life Insurance Company and Google. Additionally, Massachusetts Mutual Life Insurance Company provided support in the form of salaries for author DRD, but did not have any additional role in the study design, data collection and analysis, decision to publish, or preparation of the manuscript. The specific roles of these authors are articulated in the ‘author contributions’ section."

We note that you received funding from a commercial source: Massachusetts Mutual Life Insurance Company and Google.

"I have read the journal's policy and the authors of this manuscript have the following competing interests:

Author DRD's commercial affiliation with paid employment.

This does not alter our adherence to  PLOS ONE policies on sharing data and materials."

We note that one or more of the authors are employed by a commercial company: MassMutual Data Science.

3.1. Please provide an amended Funding Statement declaring this commercial affiliation, as well as a statement regarding the Role of Funders in your study. If the funding organization did not play a role in the study design, data collection and analysis, decision to publish, or preparation of the manuscript and only provided financial support in the form of authors' salaries and/or research materials, please review your statements relating to the author contributions, and ensure you have specifically and accurately indicated the role(s) that these authors had in your study. You can update author roles in the Author Contributions section of the online submission form.

3.2. Please also provide an updated Competing Interests Statement declaring this commercial affiliation along with any other relevant declarations relating to employment, consultancy, patents, products in development, or marketed products, etc.  

4. We note that Figures F1, S14 - S21 in your submission contain map images which may be copyrighted. All PLOS content is published under the Creative Commons Attribution License (CC BY 4.0), which means that the manuscript, images, and Supporting Information files will be freely available online, and any third party is permitted to access, download, copy, distribute, and use these materials in any way, even commercially, with proper attribution. For these reasons, we cannot publish previously copyrighted maps or satellite images created using proprietary data, such as Google software (Google Maps, Street View, and Earth). For more information, see our copyright guidelines: http://journals.plos.org/plosone/s/licenses-and-copyright.

4.1.    You may seek permission from the original copyright holder of Figures F1, S14 - S21 to publish the content specifically under the CC BY 4.0 license. 

4.2.    If you are unable to obtain permission from the original copyright holder to publish these figures under the CC BY 4.0 license or if the copyright holder’s requirements are incompatible with the CC BY 4.0 license, please either i) remove the figure or ii) supply a replacement figure that complies with the CC BY 4.0 license. Please check copyright information on all replacement figures and update the figure caption with source information. If applicable, please specify in the figure caption text when a figure is similar but not identical to the original image and is therefore for illustrative purposes only.

5. We note you have included a table to which you do not refer in the text of your manuscript. Please ensure that you refer to Tables 3, 6, 9 in your text; if accepted, production will need this reference to link the reader to the Table.

Reviewers' comments:

Reviewer's Responses to Questions

**Comments to the Author**

1. Is the manuscript technically sound, and do the data support the conclusions?

Reviewer #1: Yes

Reviewer #2: Yes

2. Has the statistical analysis been performed appropriately and rigorously? 

Reviewer #1: Yes

Reviewer #2: Yes

3. Have the authors made all data underlying the findings in their manuscript fully available?

Reviewer #1: Yes

Reviewer #2: Yes

4. Is the manuscript presented in an intelligible fashion and written in standard English?

Reviewer #1: Yes

Reviewer #2: Yes

5. Review Comments to the Author

Reviewer #1: I thank the authors for their replies to my points.There is only one thing that is not convincing me 100%. RE: spelling of "huracán", it's great to know that it was only a typesetting problem in the paper! However, I also pointed out that the accentless version of the hashtag (huracan) was more popular than either accented version. It's not super clear to me from the authors' reply that they checked also this version, which they probably should, given that it is a valuable quantity of data (although with an incorrect spelling). All in all, this should be possible to fix in a minor revision.

Reviewer #2: My comments and suggestions are fully addressed in the revision. The revised version is a much better paper than the initial submission and I'm glad that the authors were open to the suggestions. Best of luck!

6. PLOS authors have the option to publish the peer review history of their article (what does this mean?). If published, this will include your full peer review and any attached files.

Reviewer #1: No

Reviewer #2: No

---

## [Author Response · Author response to Decision Letter 0]

29 Apr 2021

We have attached a word document response including figures, but the main response text is as follows:

In response to the reviewer’s points, we have re-examined a few different accent scenarios. The reviewer suggested that the accent-less version of the hashtag is more popular than either accented version. We find this to be true for the hashtag of Hurricane Harvey. However, for Irma and Maria, we found the usage rates were almost the same. For the 2-gram version, the accented “Huracán María” is actually used more than “Huracan Maria”. In all cases the correctly accented 2-gram is used more( by an order of magnitude or two). Perhaps the reviewer was looking at hashtag data which does not separate hashtag popularity by tweet language first?  We agree with the reviewer that additional accented versions could be a valuable piece of data, but want to emphasize that all of the quantitative results in the main body of the paper are based only on English language tweets. We’ve included data from Spanish language 2-grams in Figure S8 of the supplementary materials to further investigate why Hurricane Maria’s measured attention is disproportionately low (relative to the destruction of the storm) in English tweets, and this figure helps to show that in Spanish tweets Maria receives comparable attention to other storms. 

 We appreciate the reviewer’s thoroughness, and hope to have addressed this concern.

---

## [Editor Report · Decision Letter 1]

3 May 2021

Hurricanes and hashtags: Characterizing online collective attention for natural disasters

PONE-D-21-06464R1

Dear Dr. Arnold,

We’re pleased to inform you that your manuscript has been judged scientifically suitable for publication and will be formally accepted for publication once it meets all outstanding technical requirements.

Kind regards,

Yong-Yeol Ahn, Ph.D.

Academic Editor

PLOS ONE
---

## [Editor Report · Acceptance letter]

10 May 2021

PONE-D-21-06464R1 

Hurricanes and hashtags: Characterizing online collective attention for natural disasters 

Dear Dr. Arnold:

I'm pleased to inform you that your manuscript has been deemed suitable for publication in PLOS ONE. Congratulations! Your manuscript is now with our production department. 

Kind regards, 

on behalf of

Dr. Yong-Yeol Ahn 

Academic Editor

PLOS ONE